# Film-Forming Polymers for Tooth Erosion Prevention

**DOI:** 10.3390/polym14194225

**Published:** 2022-10-09

**Authors:** Marina Gullo Augusto, Tais Scaramucci, Tiago Moreira Bastos Campos, Idalina Vieira Aoki, Nadine Schlueter, Alessandra Bühler Borges

**Affiliations:** 1Institute of Science and Technology, Department of Restorative Dentistry, São Paulo State University-UNESP, São José dos Campos 12245-000, Brazil; 2School of Dentistry, Centro Universitário de Cascavel–UNIVEL, Av. Tito Muffato, 317-Santa Cruz, Cascavel 85806-080, Brazil; 3Department of Restorative Dentistry, School of Dentistry, University of São Paulo-USP, São Paulo 12245-000, Brazil; 4Department of Physics, Aeronautical Technology Institute (ITA), São José Dos Campos 12228-900, Brazil; 5Department of Chemical Engineering, Polytechnic School, University of São Paulo-USP, São Paulo 12245-000, Brazil; 6Department of Conservative Dentistry, Periodontology and Preventive Dentistry, Hannover Medical School, 30625 Hannover, Germany

**Keywords:** dental erosion, dental enamel, polymers, sodium fluoride

## Abstract

Different agents have been proposed to prevent the progression of acid induced dental substance losses, which are called erosive tooth wear (ETW), such as fluorides, calcium, and phosphate-based products; however, there is a need for a further increase in efficacy. Recently, the ability of polymers to interact with the tooth surface, forming acid resistant films, has come into the focus of research; nevertheless, there is still the need for a better understanding of their mode of action. Thus, this article provides an overview of the chemical structure of polymers, their mode of action, as well as the effect of their incorporation into oral care products, acid beverages, and antacid formulations, targeting the prevention of ETW. Recent evidence indicates that this may be a promising approach, however, additional studies are needed to confirm their efficacy under more relevant clinical conditions that consider salivary parameters such as flow rate, composition, and clearance. The standardization of methodological procedures such as acid challenge, treatment duration, and combination with fluorides is necessary to allow further comparisons between studies. In conclusion, film-forming polymers may be a promising cost-effective approach to prevent and control erosive demineralization of the dental hard tissue.

## 1. Introduction

Over the recent decades, erosive tooth wear (ETW) has been recognized as a frequent condition, with prevalence rates reaching up to 50% of deciduous teeth and 45% of permanent teeth [1]. This condition has a multifactorial etiology, with acids from non-bacterial origin as the primary etiological factor [2]. In the early stages, erosive challenges lead to a partial loss of the superficial mineral content from the tooth, resulting in a decrease in enamel hardness [3]. With recurrent acid exposure and the association with mechanical factors, the softening is followed by a continuous layer-by-layer dissolution of enamel, leading to permanent loss of the tooth’s structure [4]. Thus, the adoption of preventive measures is crucial to avoid further functional and esthetic impairment [5].

Besides the control of causative factors, some strategies have been proposed to prevent the progression of erosive tooth wear, such as the use of products containing monovalent fluorides, e.g., sodium fluoride [6]. These products are highly recommended and widely used in oral care, in particular in the context of caries prevention. Nevertheless, as opposed to caries lesions, in which demineralization occurs in sub-surfaces areas and fluoride-driven remineralization occurs in deeper levels, in erosion, the demineralization is mainly a surface-controlled process and, therefore, comparable remineralization does not take place. Consequently, the preventive effect of monovalent fluorides is limited to the tooth surface [7,8,9], since they promote the deposition of precipitates, similar to calcium fluoride (CaF_2_), that act as a protective layer during acid episodes [6].

Products containing monovalent fluorides offer a small degree of protection against ETW. This is conceivably sufficient for most individuals with average acid exposure showing no further risk factors for the development of ETW [10,11]. However, for patients with a high risk for ETW development, a more effective product is necessary. Thus, the addition of specific compounds, such as polyvalent metal cations in the form of stannous chloride (SnCl_2_), stannous fluoride (SnF_2_), or titanium tetrafluoride (TiF_4_), to oral care products has been investigated. The metal cations can be incorporated into tooth structures, reducing enamel and dentin solubility [12,13]. Moreover, they react with the phosphate groups of the dental tissue surface or with fluoride ions from the oral care preparation, forming, in addition to CaF_2_, stable precipitates on the tooth surface (complex stannous and phosphate ion containing compounds or titanium oxides), which create layers notably more acid-resistant than pure CaF_2_ [14,15].

However, polyvalent metal cations can cause some side effects or might have limitations in use. While the use of stannous ions might be associated with some drawbacks, such as tooth discoloration and astringent feeling on the mucosa [16,17], the titanium ions form some whitish-yellowish coverages. Furthermore, the protective effect of products containing stannous or titanium ions depends on the pH [18]. In both cases, acidic preparations are more effective than neutral ones. Several studies have shown that TiF_4_ is particularly effective at pH 2, which is not an option for over-the-counter oral care products. Furthermore, in large parts of the world, titanium fluoride has no approval for use in cosmetic or medical products. Stannous ion-containing preparations, on the contrary, are highly effective at pH 4.5 and, therefore, useable in home-care products. However, the available, slightly acidic, stannous ion-containing preparations are in some cases not well tolerated, in particular if the saliva flow rate is impaired. Therefore, an increase in acceptance with concomitant increase in efficacy would be desirable.

Recently, various polymers have been investigated as agents added not only to oral care products, but also to the acid itself. It has been shown that they are able to interact with hydroxyapatite surfaces, forming acid resistant films, but there is no consensus about their overall efficacy [19]. With the aim of providing more information for the search of promising anti-erosive polymers, this article provides an overview about the chemical structure of polymers, their mode of action, as well as the effect of their incorporation into oral care products, acid beverages, and antacid formulations, targeting the prevention of ETW.

## 2. Polymers’ Structure

The word polymer originates from Greek: poly (many) and mero (parts). Polymers are organic or inorganic materials containing a structure based on the repetition of small units [20]. They can be classified as (1) natural organic polymers: naturally synthetized, such as wood, rubber, cotton, leather, silk, or even proteins, enzymes, starches, and cellulose; (2) natural inorganic polymers: naturally synthetized and found in ionic compounds and mineral salts, such as diamond or graphite; (3) artificial polymers: natural organic polymers artificially modified by chemical reactions, such as cellulose acetate, cellulose nitrate, and chitosan; (4) synthetic organic polymers: artificially synthetized, containing an organic backbone, such as polyethylene, polystyrene, and polymethacrylates; (5) synthetic inorganic polymers: artificially synthetized, containing an inorganic backbone, such as polyphosphoric acid or polyphosphates [21].

The polymerization reaction is responsible for the formation of a long polymeric molecule containing thousands of repeated units that are covalently bonded. Figure 1 shows the example of ethylene, a molecule that can be combined with a catalyst to form polyethylene, one of the most important plastic materials today. To facilitate the presentation of the polymer’s chemical structure, the repeated unit is placed in parentheses with a subscripted “*n*” that represents the polymerization degree, which is the number of times in which the units are repeated.

The control of the polymerization reaction can produce a vast range of polymers with different molecular weights, and, consequently, different properties [21], which can impact the reactivity and the formation of a polymeric film on the tooth surface. For example, previous findings showed that under erosive conditions without mechanical impacts, a high molecular weight polymer (chitosan) showed a protective effect to the enamel, but under chemo-mechanical challenges, a low molecular weight polymer behaved better [22]. This may occur because the higher molecular weight polymer created thicker precipitates, which were more able to prevent the underlying structure from coming into direct contact with the acid. By contrast, a lower molecular weight polymer implies shorter chains of molecules and, consequently, more available binding sites for the retention [22]. Further research, investigating this interaction in detail, remains necessary. One important aspect to be considered in studies is the report of the polymer’s molecular weight, allowing the comparison between them.

The control of the polymerization reaction can also produce different polymers by combining one monomer with different catalysts, producing homopolymers (Figure 2), or combining different monomers, producing copolymers (Figure 3). Thus, it is evident that the number of potential polymers is vast, but, currently, only a small part of this wide range of possibilities has been studied, and an even smaller number is commercially available.

## 3. Polymers on ETW Prevention

### 3.1. Mechanism of Action

The idea of forming a protective film on dental hard tissue to prevent the direct contact with acids is based on the function of the acquired salivary pellicle, which does this naturally [23,24]. The salivary pellicle is a thin coating formed on the dental hard tissues, consisting of precipitated and agglomerated salivary proteins. This layer directly protects the hard tissue against mechanical and chemical impacts. Moreover, the proteins from the basal pellicle layer present binding sites for calcium and phosphate, maintaining a high concentration of these ions near the hydroxyapatite surface of the dental hard tissue [25], resulting in less demineralization. The pellicle’s protective effect might be further enhanced by covering the tooth surface with additional polymers. However, in order to select suitable polymers that have the potential to form ionic bonds with hydroxyapatite and the salivary pellicle, it is necessary to understand the possible chemical interactions involved.

In a liquid medium (polar solvent), the hydroxyapatite acquires a superficial electric charge due to the electrostatic potential generated by the calcium and phosphate ions [26,27]. This surface charge will be compensated for by an equal and opposite charge of counter ions to maintain neutrality. Thus, on the interface between each hydroxyapatite particle and the surrounding liquid, an electric double layer is formed, which can be subdivided into: (1) an inner layer containing strongly bonded counter ions and (2) a diffuse layer containing loosely bounded counter ions coexisting with ions with the same charge of particle [28]. In this context, the point of zero charge (PZC) is named and corresponds to the value in which a surface is electrically neutral [20]. Previous studies indicate that the point of zero charge of the hydroxyapatite in deionized water is between five and seven [29,30]; however, it may be altered due to changes in Ca^2+^ and PO_4_^3−^ concentrations in the surrounding liquid [31]. Figure 4 shows the effect of pH changes on the surface charge of the hydroxyapatite and, consequently, on the de/remineralization reaction.

Another term related to the characterization of the electrochemical equilibrium at interfaces is the zeta potential, which corresponds to the electric potential at the shear layer of a surface in a suspension (saliva). The electric potential describes the ability of a field (caused by a charge) to exert force on other charges. Under physiological conditions in the oral cavity, the hydroxyapatite presents a predominantly negative zeta potential [26,32,33,34], which favors the ionic interaction with cationic polymers, such as some polymethacrylate copolymers [35,36] and chitosan [22]. However, in acidic pH, the relative charge changes, and the interaction with anionic polymers, such as sodium polyphosphate [33], milk casein [37], carbopol [38,39], and pectin [40], is favored.

The ionic interaction between the hydroxyapatite of the tooth surface and a polymer initiates the binding between both reactants. Besides this initial chemical interaction with the tooth surface, some polymers can additionally promote chain cross-linking through the action of divalent ions such as Ca^2+^ [41,42]. This cross-linking of polymer chains creates a gel-like structure that can cover and, consequently, protect the tooth’s surface. In this context, it is important to emphasize that the properties of this structure depend on the size (ion radius), the number of available ions, and the ease of assembly or arrangement of the polymer chains around the ions [41]. Moreover, the ionic cross-linking of some polymers, such as chitosan, can be influenced by the degree of deacetylation and/or the molecular weight [22]. Moreover, these polymers can be functionalized with a wide range of functional groups to modulate its superficial composition to a specific application [43].

### 3.2. Polymers as Active Ingredients in Oral Care Products

Technological improvements and market competition have favored the development of a wide range of oral care products for home use, whose consumption has increased significantly in the last decades [44].

By forming polymer layers on teeth surfaces, the underlying tissue can be protected against direct contact with the hydrogen ions from the acidic medium. However, a promising approach to prevent ETW seems to rely not only on the use of polymers, but also on their combination with other preventive compounds, such as fluorides. The combination can have some synergistic effects by binding of the polymer to phosphate or calcium sites of hydroxyapatite, where fluoride was not bonded [38]. Moreover, some polymers might also directly interact with the fluoride ion or the counter ion, in particular, in the case of polyvalent metal cations or other active agents in the oral hygiene product, increasing their bioavailability, retention on the tooth surface, or substantivity in the oral cavity [45].

Studies about the anti-erosive effect of toothpastes containing polymers are still scarce, but they indicate that it is a very promising approach. Most of them investigated the effect of chitosan in combination with stannous and fluoride ions. This combination was able to significantly reduce the surface loss caused by erosive/abrasive episodes [46,47,48] by increasing the retention of stannous ions on the dental hard tissue. The presence of a polymethacrylate copolymer in a toothpaste formulation contributed to its adhesion to the tooth surface, improving the efficacy of the calcium phosphate to promote enamel acid resistance [35]. Furthermore, the addition of a copolymer of maleic anhydride with methyl vinyl ether combined with lactate to a NaF toothpaste has shown to improve fluoride uptake and reduce enamel solubility [49]. In vitro investigations about the supplementation of fluoride toothpastes with sodium hexametaphosphate [50] and trimetaphosphate [51,52] have also found favorable results against enamel erosion in some cases; however, some studies showed opposite results for hexametaphosphate. It is discussed that this molecule might be able to bind calcium in a stable complex and remove it from the equilibrium [53].

Another strategy for erosion prevention is the addition of casein phosphopeptide-amorphous calcium phosphate (CPP-ACP) to mousses [54,55], gums [56,57], and varnishes [58]. CPP-ACP is a compound derived from casein, a milk protein that may also be classified as a natural polymer. It acts as a buffer providing free calcium and phosphate ions and maintaining a state of supersaturation with tooth enamel, preventing demineralization [59]. However, CPP-ACP appears to be less effective than fluorides and other active agents, such as polyvalent metal cations, in the control of ETW [60,61].

Compared to studies on polymer-containing toothpastes, a higher number of studies investigating the effect of polymer additives to mouth rinses can be found in the literature, possibly due to its simpler composition and preparation in comparison to toothpastes. However, the estimation of the polymers’ effects is intricate, as the variability in the experimental parameters used is huge, as shown in Table 1. A standardization of study designs is desirable.

**Table 1 polymers-14-04225-t001:** Compilation of studies investigating the anti-erosive effect of polymer-based solutions (ordered by publication date).

Study	Type	Substrate	Acid Challenge	Intermittent Storage of Samples	Anti-Erosive Treatment				Polymer Effect
					Polymers Tested	Concentration	Duration	Association with Fluorides	
Augusto et al., 2021 [62]	In vitro	Enamel	0.3% citric acid–pH 2.6 (5 min, 4×/day, 5 days)	Human saliva	Aminomethacrylate copolymer (AMC)	20 g/L	2 min, 2×/day, 5 days	225 ppm F^−^ (NaF); 225 ppm F^−^ (NaF) + 800 ppm Sn^2+^ (SnCl_2_)	AMC has potential to enhance the anti-erosive effect of fluoride solutions.
Luka et al., 2021 [63]	In vitro	Enamel	0,5% citric acid–pH 2.4 (2 min, 6×/day, 10 days)	Mineral salt solution	Chitosan with different viscosities (50 mPas, 500 mPas)	5 g/L	2 min, 2×/day, 10 days	500 ppm F^−^ (AmF) + 800 ppm Sn^2+^ (SnCl_2_)	Chitosan and F/Sn solution was able to reduce the tissue loss under erosive and under erosive– abrasive conditions.
Sakae et al., 2020 [64]	In situ	Enamel	1% citric acid–pH 2.3 (5 min, 4×/day, 5 days)	Human saliva	Propylene glycol alginate (PGA)	1 g/L	2 min, 2×/day, 5 days	225 ppm F^−^ (NaF);	PGA was not able to improve the protective effect of NaF against erosive enamel wear.
Souza et al., 2020 [65]	In vitro	Dentin	0,1% citric acid–pH 2,5 (90 s, 4×/day, 7 days)	Mineral salt solution	Chitosan with different viscosities (500 mPas, 2000 mPas)	5 g/L	30 s, 1×/day, 7 days	190 ppm F^−^ (NaF); 300 ppm F^−^ (NaF) + 190 ppm Ti^4+^ (TiF_4_)	Only chitosan 500 mPas was able to reduce dentin loss compared to the negative control. TiF_4_/NaF, whether with or without chitosan, had no protective effect.
Avila et al., 2020 [39]	In situ	Enamel	1% citric acid–pH 2.3 (5 min, 4×/day, 5 days)	Human saliva	Carbopol 980	1 g/L	1 min, 2×/day, 5 days	225 ppm F^−^ (NaF); 225 ppm F^−^ (NaF) + 800 ppm Sn^2+^ (SnCl_2_)	The association of Carbopol to fluoride and stannous (FS) significantly protected the enamel against erosive wear, but it was not significantly superior to FS only.
Bezerra et al., 2019 [66]	In vitro	Enamel and dentin	0.3% citric acid–pH 2.6 (5 min, 4×/day, 5 days)	Human saliva	Gantrez MS-955 Plasdone K-29/32 PGA: Propylene glycol alginate CMC: Carboxymethylcellulose	1 g/L	2 min, 2×/day, 5 days	225 ppm F^−^ (NaF); 225 ppm F^−^ (NaF) + 800 ppm Sn^2+^ (SnCl_2_)	For enamel, Gantrez, Plasdone, and CMC exhibited an anti-erosive effect, and PGA increased the protection of NaF. For dentin, only Gantrez reduced erosion.
Beltrame et al. 2018 [67]	In vitro	Dentin	0.5% citric acid–pH 2.3 (2 min, 6×/day, 5 days)	Mineral salt solution	Phosphorylated chitosan	5 g/L	2 min, 6×/day, 5 days	No	The treatment reduced erosive wear by approximately 32% in neutral and alkaline pH, when compared to the negative control.
Avila et al., 2017 [38]	In vitro	Enamel	0.3% citric acid–pH 2.6 (2 min, 6×/day, 6 days)	Mineral salt solution	Carbopol 980 Carboxymethylcellulose Aristoflex AVC	1 g/L	1 min, 6×/day, 5 days	900 ppm F^−^ (NaF)	Carbopol 980 reduced the erosive wear magnitude to the same extent as the sodium fluoride.
João-Souza et al., 2017 [68]	In situ	Enamel	1% citric acid–pH 2.4 (2 min, 6×/day, 5 days)	Human saliva	LPP: Sodium linear polyphosphate	20 g/L	2 min, 2×/day, 5 days	225 ppm F^−^ (NaF) + 800 ppm Sn^2+^ (SnCl_2_)	The presence of LPP did not enhance the anti-erosive effect of the fluoridated solution.
Pini et al., 2016 [22]	In vitro	Enamel	0.5% citric acid–pH 2.8 (2 min, 6×/day, 10 days)	Mineral salt solution	Chitosan with different molecular weight (150, 350, 400, 450 kDa)	5 g/L	2 min, 2×/day, 10 days	500 ppm F^−^ (AmF) + 800 ppm Sn^2+^ (SnCl_2_)	Under erosive conditions, the 450 kDa chitosan completely inhibited tissue loss, whereas under abrasive/erosive challenges, the 150 and 350 kDa chitosan showed the best performance, reducing by ~60% the erosive wear compared to the negative control.
Scaramucci et al., 2016 [69]	In vitro	Enamel and dentin	1% citric acid–pH 2.4 (5 min, 6×/day, 5 days)	Human saliva	Sodium linear polyphosphate	20 g/L	2 min, 3×/day, 5 days	225 ppm F^−^ (NaF); 225 ppm F^−^ (NaF) + 800 ppm Sn^2+^ (SnCl_2_)	The combination of NaF with SnCl_2_ and/or LPP can protect the enamel against erosion. For dentin, neither toothbrushing nor the test solutions influenced the development of erosion.
Scaramucci et al., 2015 [33]	In vitro	Enamel and dentin	0.3% citric acid–pH 3.8 and 1% citric acid–pH 2.4 (5 min, 6×/day, 5 days)	Mineral salt solution	Linear sodium polyphosphate Sodium pyrophosphate tetrabasic Sodium tripolyphosphate Sodium caseinate Bovine serum albumin	2 g/L and 20 g/L	2 min, 3×/day, 5 days	225 ppm F^−^ (NaF); 225 ppm F^−^ (NaF) + 800 ppm Sn^2+^ (SnCl_2_)	The addition of LPP and/or SnCl_2_ can improve the fluoride solution’s protection against erosion of enamel but not of dentine.
Lei et al., 2014 [70]	In vitro	Enamel	1% citric acid–pH 3.8 (5 min, 1×, 1 day)	No	Synthetic amphiphilic diblock copolymer	0.06, 0.12, 0.25, 0.5 and 1 g/L	5 min, 1×	No	The treatment with the polymer decreased the mineral loss of hydroxyapatite by 36–46% compared to the untreated control and protected the surface morphology of the enamel specimen following exposure to acid.
White et al., 2011 [37]	In vitro	Enamel	0.3% citric acid–pH 3.2 (10 min, 9×, 1 day)	No	Casein CPP: Casein phosphopeptide GMP: Glycomacropeptide	5 g/L	10 min, 1×	300 ppm F^−^ (NaF)	Casein and NaF reduced enamel surface softening compared to the negative control, but CPP and GMP did not.
Gracia et al., 2010 [71]	In vitro	Enamel	1% citric acid–pH 3.8 (5 min, 1×, 1 day)	No	Combination of 0.20% carboxymethylcellulose, 0.010% xanthan gum and 0.75% copovidone	-	1 min, 1×	300 mg/L fluoride	The treatment with the polymer significantly reduced the lesion depth and enhanced the delivery of fluoride to the surface of the lesion.

In most of the studies, promising results were obtained with the polymers. However, the different experimental parameters used can influence the outcome and, thus, need to be considered carefully. Based on this overview, the following variables influence the outcome and should be standardized as best as possible in further studies investigating the addition of polymers to solutions. The following list of considerations does not claim to be complete. It should be understood as a suggestion or a recommendation rather than as a standard design.

Substrate: As the anti-erosive effect of the polymers can vary on enamel and dentin, and dentin is often exposed even at early stages, both substrates should be investigated. Although bovine teeth can be used as a substitute for human teeth, caution must be taken to extrapolate the results, since there are differences regarding susceptibility to demineralization processes and interaction with active agents [72,73]. Thus, if possible and ethically justifiable, human teeth should be preferred.Acid challenge: Higher concentrations of citric acid (1% or 0.5%) may be preferred over lower ones (0.3%) [33], as a better discrimination of results is possible [53].Intermittent storage of samples: The efficacy of anti-erosive agents can be affected by protein interactions; thus, the use of human saliva is meaningful if the clinical effect is under investigation [74]. In case of evaluation of the direct interaction between the inorganic compounds of the dental hard tissue and the biopolymer, the inclusion of human saliva might hamper analysis [63].Anti-erosive treatment: As the binding affinity seems to be dependent on the polymer’s molecular weight [22], concentration, and pH of the polymer solution or the demineralization solution [70], these parameters should only be changed if it is the intention of the study to investigate their impact; otherwise, these parameters should be kept constant in order to allow comparability.Duration of the treatment: a representative period for testing solutions simulating mouth rinses should preferably not exceed the clinical condition. This means approximately 1–2 min for mouth rinses under in vitro conditions [75] and 30 s under in situ conditions [76].Association to fluorides: Since fluorides remain the standard in erosion prevention, and polymers added to oral care products may complement the protective effect of fluorides, the interaction between the efficacy of fluorides and polymers should be addressed.

### 3.3. Polymers as Active Ingredients in Acidic Beverages

In addition to symptomatic strategies, in which polymers are used to inhibit the interaction between acid and dental hard tissue, causal strategies, in which the erosivity of the acid itself is modified, also appear to be useful. Attempts to reduce the erosive potential of acidic beverages by supplementation with mineral agents have been reported [77]; however, although successful, this approach presents some drawbacks, mainly related to taste alterations, stability difficulties, and toxicity potential [78,79,80,81]. The use of food-approved polymers is a promising alternative to overcome these limitations [82].

Regarding the effect of food-added polymers, different modes of action are conceivable. It can be expected that the polymers directly reduce erosive demineralization by adsorbing on the tooth surfaces, creating a protective film comparable to the mode of action described for the oral hygiene products. Furthermore, the polymers can also increase the viscosity of acidic drinks, reducing the ion mobility in the solution and, consequently, slowing down the dissolution kinetics [40]. Likewise described for the polymers’ effects in oral hygiene products, their efficacy as a food additive is related to the polymer type, and contradictory results have been reported [77,83].

The erosive potential of acid beverages was efficiently reduced by its modification with polymers under in vitro conditions. The ability of some food-approved condensed phosphates and gums in reducing hydroxyapatite dissolution caused by a standard citric acid solution was previously tested. The compounds were added at 0.02% concentration. Linear sodium polyphosphate (LPP) reduced hydroxyapatite dissolution by 64%, sodium tripolyphosphate by 46%, and sodium pyrophosphate by 35%. The superiority of linear sodium polyphosphate was attributed to its longer chain length, which promoted a better coverage of the surface. Among the gums, xanthan gum was the most promising, but promoted a more discrete reduction in hydroxyapatite dissolution (29%) [82].

In another study, these compounds were added to an orange juice at 0.02% concentration, and linear sodium polyphosphate was the most promising, reducing hydroxyapatite dissolution by about 84%. In this same study, orange juice modified with linear sodium polyphosphate was also able to reduce enamel loss by erosion significantly in comparison to the non-modified juice. Stimulated by these promising results, the linear sodium polyphosphate-modified orange juice was tested using an in situ model. However, its anti-erosive effect was not significantly different from the original juice. Under this condition, erosion protection was only observed when linear sodium polyphosphate was combined with calcium. It was suggested that a higher concentration of the polymer is needed in situ than in vitro, once in situ the polymers must face the competition for binding sites at the dental surfaces with the proteins of the salivary pellicle [84]. In contrast, in another in situ trial, a sodium polyphosphate was tested as an additive of a soft drink at 0.02% concentration. The modified soft drink reduced erosion significantly in comparison to the original soft drink (in the order of 94%) [85]. Xanthan gum was also investigated, but was not able to further reduce the anti-erosive effect of a calcium-modified blackcurrant drink; nonetheless, it was argued that its addition helped to improve the stability and taste of the drink [86].

Other polymers were also able to successfully reduce the anti-erosive effect of citric acid in vitro, such as highly esterified pectin, propylene glycol alginate, and gum arabic, at 1% concentration. These anionic polymers showed different abilities to prevent the loss of enamel nanohardness, depending on the pH of the citric acid, which was tested in the range of 2.2 to 4.0 [40]. For highly esterified pectin and propylene glycol alginate, the interaction occurred between the carboxyl groups of the polymers with the calcium sites at the enamel surface. For gum arabic, it was described that the polymer promoted electrostatic and hydrogen bonds with the positively charged enamel surface sites.

Proteins can be considered natural polymers in which the amino acids are linked by amide bonds [87]. Casein is a milk-derived protein, and ovalbumin is derived from egg white. They both contain phosphoserine sequences which can bind to hydroxyapatite [88,89] and, potentially, to the salivary pellicle [90]. Casein at 0.02% reduced hydroxyapatite dissolution by about 50% when added to citric acid solutions with different pH values. Higher concentrations did not result in further protection, but combining casein with calcium (calcium concentrations varying from 5 to 10 mM) enhanced the anti-erosive effect [91]. Casein and ovalbumin were then tested as additives of pure acidic solutions and some acid drinks regarding their ability to reduce enamel erosion in the presence of the salivary pellicle. In citric acid, 0.2% ovalbumin reduced erosion in the range of 46–61%, depending on the pH (2.8–3.8), whereas 0.2% casein reduced erosion in the range of 59–78%, following the same pH range. In soft drinks, the effect was lower, with an average numerical reduction in dissolution by 20% with ovalbumin and 36% with casein [90]. In another in vitro study, 0.2% casein and 2% ovalbumin were added to an orange juice. The juices were tested on enamel and dentin. On enamel, both proteins did not differ significantly, but they reduced surface loss by erosion in about 60% when compared to the original juice. In dentin, by contrast, the proteins did not significantly reduce the anti-erosive effect of the juice [92].

Given the vast range of food-approved polymers that are available, it is clear that this is a field that needs to be better explored, especially when considering the usefulness of such beverages in the cases where the patients have difficulties in complying with the treatment. It should be kept in mind, however, that the polymers can also alter the taste and visual appearance of the drink (e.g., milky turbidity). Additionally, factors such as viscosity, stability, and acceptance by consumers also need to be considered.

### 3.4. Polymers as Active Ingredients in Antacid Formulations

Erosive tooth wear cannot only be induced by acids from nutrition but also by intrinsic acids. The only intrinsic acid is gastric acid, which can reach the oral cavity during reflux episodes or vomiting. Patients with such diseases as gastric reflux or bulimia nervosa commonly use antacid formulations. The pharmaceutical industry has added polymers to these products in order to enhance the retention of antacid formulations within the stomach, resulting in a longer-lasting effect of neutralization compared to conventional antacids [93].

Alginate is a natural polysaccharide composed of α-d-mannuronic acid and β-l-guluronic acid. This polymer forms gel in the presence of various divalent cations, e.g., Ca^2+^, Mg^2+^, or Al^3+^, by cross-linking the carboxylate groups of the guluronate groups on the polymer backbone [94]. An antacid containing alginate was shown to suppress reflux after meals by creating a gel-like barrier that hampers the acid from passing the esophago-gastric junction [95]. In this context, in vitro findings have shown that rinsing with antacid suspension containing sodium alginate, sodium bicarbonate, and calcium carbonate was effective to decrease signs of erosion measured by both an enamel microhardness reduction and surface loss. This effect could be related to viscoelastic polymer chains formed by the alginate on the tooth’s surface [96,97].

Moreover, future studies should investigate whether patients using medications such as a conventional antacid, and not a mouth rinse, show a clinically lower prevalence of ETW, since the polymer may reduce the ion mobility of the hydrochloric acid in the stomach.

## 4. Conclusions and Future Perspectives

(a) Film-forming polymers may represent a promising cost-effective approach to prevent and control erosive demineralization of the dental hard tissue.

(b) Additional studies are needed to confirm their efficacy under more relevant clinical conditions considering salivary parameters such as flow rate, composition, and clearance. The influence of the acquired salivary pellicle should also be part of these studies.

(c) The standardization of study conditions is necessary to allow comparisons among studies.

## Figures and Tables

**Figure 1 polymers-14-04225-f001:**
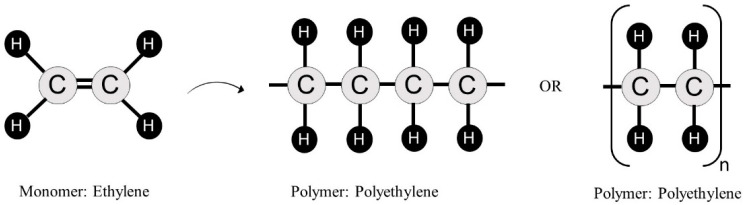
Illustration of the ethylene polymerization.

**Figure 2 polymers-14-04225-f002:**
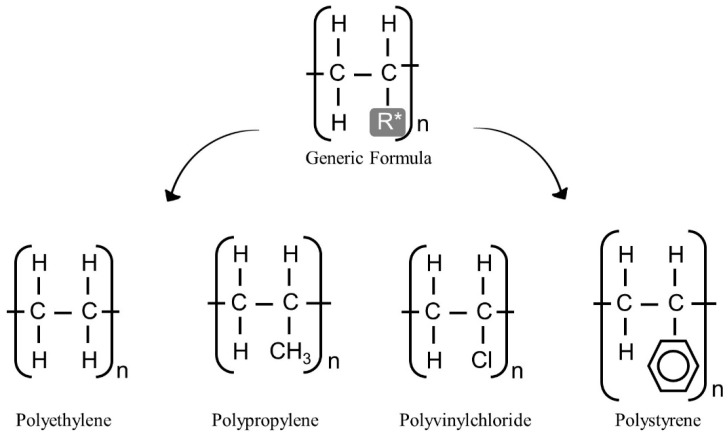
Representation of polymers that share the same generic formula. R*: Free radical.

**Figure 3 polymers-14-04225-f003:**
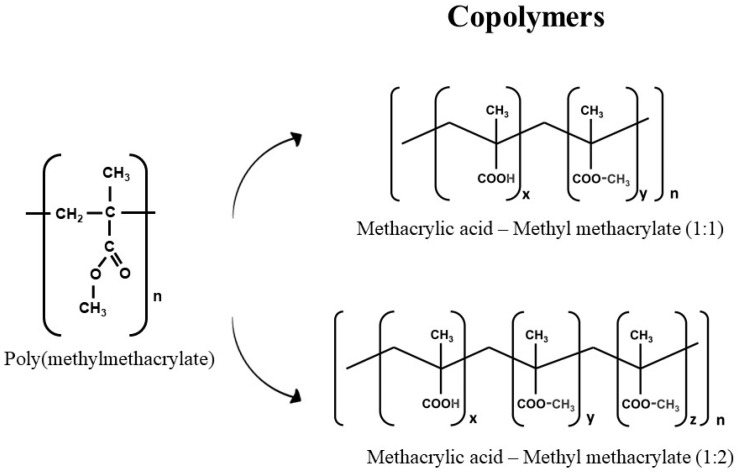
Representation of the PMMA chemical structure and two copolymers.

**Figure 4 polymers-14-04225-f004:**
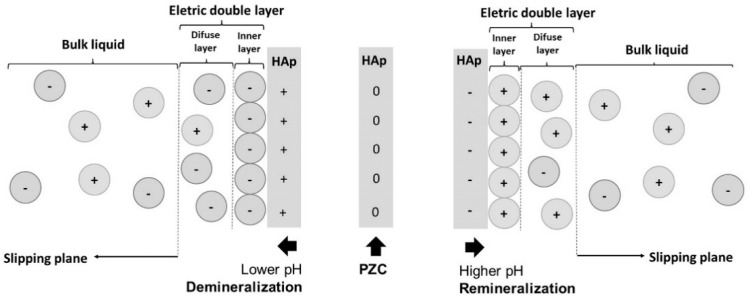
Representation of the electric double layer on the hydroxyapatite surface.

## Data Availability

Not applicable.

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
