# Peer review of "Film-Forming Polymers for Tooth Erosion Prevention"

_polymers, 2022, doi:10.3390/polym14194225_

Round 1

Reviewer 1 Report

This manuscript has great innovative significance in investigating film-forming polymers for tooth erosion prevention. The work can arouse wide interests of researchers in design and preparation of new functional materials. The manuscript is interesting. In my frank opinion, the manuscript should be deserved for its final publication in such high-level Journal. The main reasons are as follows:

1. At first, the English ABSTRACT should be revised, and a unified simple present tense should be used.

2. The research significance and future work should be described in the final stage of the abstract.

3. Aims need to be concisely stated and added at the end of introduction. Not only what was done/investigated, but why.

4. At the same time, pay attention to some presentation styles, such as suggesting to avoid direct use of overlong sentences, and pay attention to the logic of the use of conjunctions.

5. Under normal conditions, in conclusion section, important conclusions should be elaborated point by point for brevity and prominence, such as a) … … b) … … c) … ….

6. And also in the last point future research work should be given in conclusion section.

Author Response

We would like to thank the reviewer for the time and effort for the appraisal of the manuscript. All your comments were addressed. The manuscript was overall reviewed for improving understanding.

The abstract was reviewed. The following sentences describe the research significance and future work in abstract: “Recent evidences indicate that this may be a promising approach; however, additional studies are needed to confirm their efficacy under more relevant clinical conditions, considering salivary parameters, such as flow rate, composition, and clearance. The standardization of methodological procedures as acid challenge, treatment duration, and combination with fluorides is necessary to allow further comparisons between studies.”

The introduction was reviewed according to the suggestions, attempting to better describe the relevance of the review: “Recently, various polymers have been investigated as protective agents added not only to oral care products, but also to the acid itself, as an alternative to reduce the acid erosive potential. It has been shown that they are able to interact with hydroxyapatite surfaces forming acid resistant films, but there is no consensus about their overall efficacy. The protective effect of fluoride against tooth demineralization can also be improved by combining this agent with bioadhesive polymers [21]. Nevertheless, there still a need to further understand the different structure of polymers and their potential interaction with the tooth surface, as well as with other active agents present in oral care formulations. Additionally, the feasibility of its clinical use should be addressed. With the aim of providing more information for the search of promising and effective anti-erosive polymers, this article provides an overview about the chemical structure of polymers, their mode of action, as well as the effect of their incorporation to oral care products, acid beverages, and antacid formulations, targeting the prevention of ETW.”

The conclusion was reformulated, as suggested.

Reviewer 2 Report

Dear Authors,

thank you for your work which I found really relevant.

However, I think that much recent literature is missing in particular relative to biomimetic hydroxyapatite .

I think you should refer to the studies of Butera, as the following:

https://www.mdpi.com/2073-4360/13/16/2740

I thus suggest consideration after major revisions. 

Yours faithfully 

Author Response

Thank you very much for taking the time and efforts for the appraisal of the manuscript. Although we agree that the studies using biomimetic hydroxyapatite are relevant and recent, such materials represent a distinct strategy to prevent/control tooth demineralization. The Butera et al. study evaluated the deposition of ions on the surfaces of bulk-filled composite resins after the use a toothpaste containing Zn-hydroxyapatite and did not fit in our theme. Thus, unfortunately, we could not include the suggested reference. We, however, have saved the reference to be used in future studies.

Reviewer 3 Report

Current manuscript entitled “Film-forming polymers for tooth erosion prevention” by “Augusto et al” deliberated on the polymers for preventing the tooth erosion. Manuscript seems good and interesting. However, the authors should address the following comments.

1.     Revise the abstract, its not giving clear idea.

2.     Introduction seems very less. Discuss more on the current topic, its background and its importance.

3.     Conclusions is not enough, provide adequate conclusions.

4.     Provide the challenges that are currently facing with the erosive tooth wear.

Author Response

Dear reviewer, thank you very much for the time and efforts to review our study. The manuscript was overall reviewed, and all your comments were addressed. The abstract and introduction were reviewed, as suggested.

The conclusion was reformulated, including a more comprehensive and broader perception of the addressed review topic, as well as future perspectives.